# The C-Terminal Region of SLIM1 Transcription Factor Is Required for Sulfur Deficiency Response

**DOI:** 10.3390/plants11192595

**Published:** 2022-10-02

**Authors:** Justyna Piotrowska, Yuki Jodoi, Nguyen Ha Trang, Anna Wawrzynska, Hideki Takahashi, Agnieszka Sirko, Akiko Maruyama-Nakashita

**Affiliations:** 1Institute of Biochemistry and Biophysics Polish Academy of Sciences, ul. Pawinskiego 5A, 02-106 Warsaw, Poland; 2Department of Bioscience and Biotechnology, Graduate School of Bioresource and Bioenvironmental Sciences, Faculty of Agriculture, Kyushu University, 744 Motooka, Nishi-ku, Fukuoka 819-0395, Fukuoka, Japan; 3RIKEN Plant Science Center, 1-7-22 Suehiro-cho, Tsurumi-ku, Yokohama 230-0045, Kanagawa, Japan; 4Department of Biochemistry and Molecular Biology, Michigan State University, East Lansing, MI 48824, USA

**Keywords:** SLIM1 transcription factor, sulfur deficiency, *Arabidopsis thaliana*, sulfate transporter, sulfate assimilation

## Abstract

Sulfur LIMitation1 (SLIM1) transcription factor coordinates gene expression in plants in response to sulfur deficiency (−S). SLIM1 belongs to the family of plant-specific EIL transcription factors with EIN3 and EIL1, which regulate the ethylene-responsive gene expression. The EIL domains consist of DNA binding and dimerization domains highly conserved among EIL family members, while the N- and C-terminal regions are structurally variable and postulated to have regulatory roles in this protein family, such that the EIN3 C-terminal region is essential for its ethylene-responsive activation. In this study, we focused on the roles of the SLIM1 C-terminal region. We examined the transactivation activity of the full-length and the truncated SLIM1 in yeast and *Arabidopsis*. The full-length SLIM1 and the truncated form of SLIM1 with a deletion of C-terminal 106 amino acids (ΔC105) transactivated the reporter gene expression in yeast when they were fused to the GAL4 DNA binding domain, whereas the deletion of additional 15 amino acids to remove the C-terminal 120 amino acids (ΔC120) eliminated such an activity, identifying the necessity of that 15-amino-acid segment for transactivation. In the *Arabidopsis slim1-2* mutant, the transcript levels of *SULTR1;2* sulfate transporter and the GFP expression derived from the *SULTR1;2* promoter-GFP (*P_SULTR1;2_-GFP*) transgene construct were restored under −S by introducing the full-length SLIM1, but not with the C-terminal truncated forms ΔC105 and ΔC57. Furthermore, the transcript levels of −S-responsive genes were restored concomitantly with an increase in glutathione accumulation in the complementing lines with the full-length SLIM1 but not with ΔC57. The C-terminal 57 amino acids of SLIM1 were also shown to be necessary for transactivation of a −S-inducible gene, *SHM7/MSA1*, in a transient expression system using the *SHM7/MSA1* promoter-GUS as a reporter. These findings suggest that the C-terminal region is essential for the SLIM1 activity.

## 1. Introduction

Sulfur (S) is an essential nutrient for all organisms. Plants use sulfate as an S source to assimilate it into cysteine and synthesize various organic S compounds [1,2]. The importance of plants in the global S cycle in nature is evident because animals cannot assimilate sulfate and they rather consume S-containing amino acids and proteins as dietary S sources [1,2].

SLIM1/EIL3 is a transcription factor (TF) that controls the S deficiency (−S) responsive gene expression, while it belongs to the plant-specific EIL family of TFs (EIL-TFs) [3,4]. The EIL domain is conserved among the EIL-TFs, but the N- and C-terminal regions are variable (Figure 1) [5]. The best-characterized member of the EIL-TFs family is EIN3 (Ethylene-INsensitive3) which initiates downstream transcriptional cascades for ethylene responses [6,7,8]. Studies of ethylene-responsive elements demonstrated that EIN3 binds to the upstream sequence in an ethylene-responsive gene, *ERF1*, through the DNA binding domains, BD I to IV (1 to 359 amino acids of EIN3) [7]. The DNA binding domain was further narrowed down to a specific region from 174 to 306 amino acids (aa) containing BD III and IV [9]. Despite that the N-terminal half region of EIN3 controls the DNA binding and its dimerization, the protein stability of EIN3 is controlled via its C-terminal region, which interacts with EBF1 (EIN3-Binding F-box Protein 1) and EBF2, responding to ethylene and the plant carbon status [7,10,11,12].

The domain structure for SLIM1 has been studied concurrently (Figure 1). Like EIN3, the N-terminal region of SLIM1 (75 to 286 aa), including BD II, III, and IV, can bind to the UPE-box (AGATACATTGAACCTGGACA), which has been shown as the conserved sequence among several −S responsive genes [13,14]. The 162-to-288 aa region of SLIM1, including BD III and IV, can interact with the conserved EIL binding sequence, AYGWAYCT [7,15], although with less affinity compared to EIN3 [16]. The *slim1* mutations in the protein coding region are only found in the conserved region spanning BD II, III, and IV [3]. Despite the genetic evidence and information that supports specific DNA binding capabilities of SLIM1, molecular mechanisms involved in −S responses via the regulation of the SLIM1 protein activity has not been elucidated yet. 

The activity of SLIM1 seems to be controlled post-transcriptionally based on the observation that SLIM1 transcript levels did not increase under −S, and the complementation of *slim1* mutants only occurred under −S even though the expression of SLIM1 was driven by a cauliflower mosaic virus (CaMV) 35S promoter which allows constitutive expression [3]. Recently, it has been demonstrated that SLIM1 undergoes proteasomal degradation facilitated by the N-terminal 285 aa region [17]. In contrast to EIN3, SLIM1 interacts only with EBF1 but not EBF2, and the protein region to interact with EBF1 has been shown to be localized at the C-terminal portion (residues 287-428 of SLIM1) [17]. Further characterization of proteasomal degradation through dissection of the C-terminal region may clarify mechanisms that control the SLIM1 protein activity.

In this study, we focused on investigating the role of the SLIM1 C-terminal region by dissecting the region required for transactivation of target gene expression under −S. Our results indicated that the C-terminal region of SLIM1 is indispensable to its activity under −S.

## 2. Results and Discussion

### 2.1. The C-Terminal Region of SLIM1 Is Required for Transactivation in Yeast

To explore the function of the C-terminal region of SLIM1, the full-length and the C-terminal truncated forms of SLIM1 were expressed as fusion proteins with the DNA binding domain of GAL4 (BD-GAL4) in a histidine (His) auxotrophic yeast strain AH109, which allowed us to characterize the ability of these BD-GAL4 fusion proteins to bind to the upstream activation sequence and transactivate the *HIS3* reporter gene by monitoring the growth on the selective medium lacking His (Figure 2). For preparation of the C-terminal truncation of SLIM1, 57, 105, 120, 137, 150, and 165 aa were deleted from the C-terminal end of SLIM1 and named ΔC57, ΔC105, ΔC120, ΔC137, ΔC150, and ΔC165, respectively. Each plasmid construct with the BD-GAL4 fusion for the full-length or the C-terminal truncated SLIM1 was integrated into the yeast strain AH109 and grew on agar plates lacking tryptophan (−Trp) or Trp and His (−Trp/−His) supplemented with or without 3-AT (Figure 2). The yeast cells expressing the BD-GAL4 full-length SLIM1 fusion protein grew well on both the control (−Trp) and selective (−Trp/−His, −Trp/−His 10 mM 3-AT) media, while the empty plasmid with the BD-GAL4 alone could grow only on the control medium (−Trp). Among the yeast cells expressing the BD-GAL4 fusion proteins with the C-terminal truncated forms of SLIM1, only the ΔC57 and ΔC105 variants grew well on both the control and selective media, but others with longer deletions did not grow on the selective medium. These results indicated that the 15 aa (VNEQTMMPVDERPML) located at the positions 448 to 462 aa of SLIM1 (105 to 119 from the C-terminal end of SLIM1) are necessary for transactivation of the *HIS3* gene expression in yeast.

### 2.2. C-Terminal 57 Amino Acids in SLIM1 Are Necessary for slim1 Complementation

To further investigate the roles of the SLIM1 C-terminal region in plants, the full-length and the C-terminal truncated forms of SLIM1 were expressed in *Arabidopsis slim1-2* mutant, which allows monitoring of plant −S responses displayed as fluorescence of GFP expressed under the control of the promoter region of *SULTR1;2* sulfate transporter [3,18].

In plants exposed to −S, both the GFP fluorescence and the *SULTR1;2* transcript expression recovered by expression of the full-length SLIM1 in the *slim1-2* mutant, but not with the C-terminal 57 and 105 aa truncated forms, ΔC57 and ΔC105 (Figure 3; S0 condition). However, consistent with our previous observation [3], the recovery of these phenotypes was only seen under −S, although the CaMV 35S promoter was used for SLIM1 overexpression. We also found that both the overexpression of ΔC57 and ΔC105 weakened the *P_SULTR1;2_-GFP* derived expression of GFP fluorescence relative to *slim1-2* (Figure 3a), as well as the *SULTR1;2* transcript expression, particularly under −S conditions (Figure 3b). We obtained the similar results with the C-terminal 165 aa truncated form, ΔC165 (data not shown).

The transcript levels of other −S-responsive genes, *BGLU28*, *SDI1*, *SULTR2;1*, and *APR3*, were similarly influenced in the complemented lines; i.e., their −S responses were restored by introducing the full-length SLIM1 but not by ΔC57 (Figure 4). Interestingly, the induction of *SULTR2;1* and *APR3* gene expression under −S were also influenced by the C-terminal 57 aa, although their −S induction was reported as SLIM1-independent [3]. These results indicate that the C-terminal 57 aa region of SLIM1 is required for complementation of the *slim1-2* mutant.

To gain further insights into the −S-responsive phenotypes affected by the C-terminal 57 aa deletion (ΔC57), we analyzed the tissue sulfate, cysteine, and GSH levels. Sulfate levels were not different among the plant lines under both S1500 (S-sufficient) and S0 (−S) conditions (Figure 5). Cysteine levels were similar between the parental line and *slim1-2* but slightly increased in ΔC57/*slim1-2* compared to SLIM1/*slim1-2* under both S1500 and S0 conditions. GSH levels were lower in *slim1-2* than in the parental line under S1500 and lower in ΔC57/*slim1-2* than in SLIM1/*slim1-2* under S0 conditions (Figure 5). These results suggest that sulfate assimilation is restored by overexpression of the full-length SLIM1 to complement *slim1-2* under −S, but not with the C-terminal truncated form ΔC57.

### 2.3. Transactivation of a −S-Inducible Gene, SHM7/MSA1, Requires C-Terminal 57 Amino Acids of SLIM1

Next we tested how SLIM1 activity is affected by the C-terminal deletion using the transient *in planta* transactivation test (Figure 6). For this purpose, we selected one of the −S-inducible and SLIM1-dependent genes, *SHM7/MSA1* [3,19]. We constructed the reporter plasmid containing an *SHM7/MSA1* promoter-GUS expression cassette to monitor the transactivation of the *SHM7/MSA1* promoter by assessing the GUS activity. The reporter plasmid was transfected with the effector plasmids expressing either the full-length SLIM1 (SLIM1), or the C-terminal 57-aa truncated SLIM1 (ΔC57) under the control of CaMV 35S promoter in *Nicotiana benthamiana* leaves. The GUS activity was analyzed after 3 days of transfection (Figure 6).

The full-length SLIM1 increased the GUS activity compared to the leaf part, where only the reporter plasmid was introduced alone (Leaf 1 in Figure 6). In contrast, ΔC57 did not show such an increase after the infiltration (Leaf 2 in Figure 6). When compared between the full-length SLIM1 and ΔC57, the GUS activity was highly increased with the full-length SLIM1 infiltration (Leaf 3 in Figure 6). These results indicate that the C-terminal 57 aa region of SLIM1 is essential for activating SLIM1-controlled genes in plants.

### 2.4. Roles of SLIM1 C-Terminal Region

In this study, we found that the C-terminal region of SLIM1 is important for its protein functionality. We demonstrated the importance of the C-terminal 57 aa in activating expression of SLIM1-responsive genes in plants by using transgenic plants expressing *P_SULTR1;2_-GFP*. Besides the importance of this C-terminal 57 aa region *in planta*, our experiments revealed the function of an additional 15 aa region located between the positions 105 to 119 aa from the C-terminal end of SLIM1 to function as a transcriptional activator in yeast. The contribution of this putative 15 aa activation domain remains an unsolved question *in planta*, because its presence in ΔC57 deletion was not sufficient to complement the *slim1* mutant. 

Domain search tools in InterPro (http://www.ebi.ac.uk/interpro/, accessed on 9 July 2022) and Conserved Domains (https://www.ncbi.nlm.nih.gov/Structure/cdd/cdd.shtml, accessed on 9 July 2022) did not detect presence of significant domains in the C-terminal 120 aa of SLIM1, indicating that our findings may lead the novel molecular machinery controlling the SLIM1 functions. The SLIM1 protein stability is controlled by the N-terminal half (1-285 aa) [17]. In contrast, the C-terminal region could play roles in other aspects implicated for SLIM1 function, such as possible interaction with other proteins or protein modification required for transactivation. It would be necessary to examine the function of C-terminal 57 aa by further dissecting this region for *slim1* mutant complementation to clarify the molecular machinery of transcriptional activation associated with SLIM1 and connect these findings with the question of how plants respond to −S.

## 3. Materials and Methods

### 3.1. Yeast Assay

Full-length or truncated *SLIM1* coding sequences were amplified with PCR using the forward primer SLIM1-F (5′-CACCATGGGCGATCTTGCTATGTCCGTAGC-3′) in combination with the reverse primer, SLIM1-R (5′-CTAAGCTCCAAACCATGAGAAATCATCACCA-3′ for full length), SLIM1-510R (5′-ACTGTTGTGAGGTGGTGCTTGTGTATTCATTTCT-3′ for truncation of C-terminal 57 aa, ΔC57), SLIM1-461R (5′-AAGCATTGGCCTTTCGTCTACAGGCATCATAGT-3′ for ΔC105), SLIM1-446R (5′-GTGGAAGATAAGTATAGTTGTTATTGAACTCAGGA-3′ for ΔC120), SLIM1-430R (5′-CAGACCATTATCCTCTGGTCCTAAGGCA-3′ for ΔC137), SLIM1-417R (5′-GACAACATCGTCCTCTTGATGAGTACCGT-3′ for ΔC150), or SLIM1-402R (5′-CAGAGGGGCATCAACATGATTCATATCAGGT-3′ for ΔC165), and KOD-Plus (TOYOBO, Osaka, Japan). The resultant fragments were purified from the agarose gel and cloned into pENTR/D-TOPO vector (Invitrogen, Waltham, MA, USA). After validating the sequence, each coding sequence was integrated into pBD-GAL4-GWRFC [20] using LR clonase (Invitrogen). The resultant plasmids were then transferred to *S. cerevisiae* strain AH109 (Clontech, Mountain View, CA, USA) according to the manufacturer’s instructions (Matchmaker GAL4 Two-Hybrid System 3, Clontech). The yeast transformants were incubated at 30 °C on minimal SD medium (Clontech) lacking tryptophan (Trp) for three days. To verify the transcriptional activation through the function of GAL4-BD fusion proteins with the full-length and the C-terminal truncated forms of SLIM1 in yeast cells, four independent colonies per each plasmid were resuspended in sterilized double-distilled water and dropped on minimal SD medium lacking Trp or Trp and His with or without 10 mM 3-AT (TCI, Tokyo, Japan). pBD-GAL4 Cam plasmid (Stratagene, San Diego, CA, USA) was used as the negative control.

### 3.2. Plant Materials and Growth Conditions

*P_SULTR1;2_-GFP* plants (Parental, *Arabidopsis thaliana* Columbia-0 (Col-0) accession) [18], *slim1-2* [3], and *slim1-2* complemented with the full-length or truncated *SLIM1* coding sequences were used as the plant samples. 

To introduce the full-length *SLIM1* (SLIM1) and truncated *SLIM1* variants named ΔC57 and ΔC105, their coding sequences cloned into pENTR/D-TOPO vector were integrated into pH35GS binary vector [21] using LR clonase (Invitrogen). The resultant binary plasmids were transferred to *Agrobacterium tumefaciens* GV3101 (pMP90) [22] and used for the transformation of *slim1-2* plants [23]. The transgenic plants were selected on GM media containing 30 mg L^−1^ hygromycin sulfate. 

The *T*_2_ progenies of SLIM1/*slim1-2*, ΔC57/*slim1-2*, and ΔC105/*slim1-2* transgenic lines, and the parental line and *slim1-2* plants were vertically grown for 10 days on the agar medium [24] supplied with 1500 µM (S1500), 15 µM (S15), or 0 µM (S0) sulfate at 22 °C and 16 h/8 h light (40 µmol m^−2^ s^−1^) and dark cycles.

### 3.3. Imaging of GFP Fluorescence

The expression of GFP in the whole intact seedlings was visualized by using the image analyzer, Amersham Typhoon scanner 5, equipped with a 525BP20 filter and a 488-nm laser (GE Healthcare, Chicago, IL, USA) as described previously [25].

### 3.4. Quantitative RT-PCR

Total RNA was extracted from root tissues using Sepasol-RNA I (Nacalai Tesque, Kyoto, Tapan) and reverse transcription was conducted using the PrimeScript RT Reagent Kit with gDNA Eraser (Takara, Japan). Subsequently, quantitative PCR was conducted using SYBR *Premix Ex Taq* II (Takara, Japan) and a Thermal Cycler Dice Real Time System (Takara, Japan). Relative mRNA abundance was calculated using ubiquitin (UBQ2, accession no. J05508) as a constitutive internal control. Gene-specific primers for quantitative PCR were previously described [3,26,27].

### 3.5. Measurements of Sulfate and Thiols (Cysteine and GSH)

Plant tissues were flash frozen in liquid nitrogen and extracted with 5 volumes of 10 mM HCl by homogenizing with Tissue Lyser MM300 (Retsch, Haan, Germany). The resultant mixtures were centrifuged at 4 °C, 13,000 rpm for 15 min, and the supernatant was used for sulfate and thiols analyses.

Sulfate contents were determined by ion chromatography (IC-2001, TOSOH, Tokyo, Japan) as described previously [28]. Cysteine and GSH contents were determined by HPLC-fluorescent detection system after labeling of thiol bases by monobromobimane as described previously [27]. The labeled products were separated by HPLC using the TSKgel ODS-120T column (150 × 4.6 mm, TOSOH) and detected using a scanning fluorescence detector FP-920 (JASCO, Oklahoma City, OK, USA), monitoring for fluorescence of thiol-bimane adducts at 478 nm under excitation at 390 nm. Cysteine and GSH (Nacalai Tesque, Kyoto, Japan) were used as standards.

### 3.6. Transient Trans-Activation Assay

pH35GS-SLIM1 and pH35GS-ΔC57 were used as the effector plasmids. For the reporter plasmid, we inserted *SHM7/MSA1* promoter sequence (569 bp upstream of ATG) amplified by PCR using the *A. thaliana* genomic DNA and primers, F: 5′-CACCTACCATAGTCCAACTCCATCC-3′ and R: 5′-ACGTTGAAGATGATGAAGATTTGG-3′, into *Not*I and *Xba*I sites of the modified pGreenII0029 binary vector [29] and validated the sequence. The *trans*-activation assay was performed as previously described [13]. In brief, the reporter plasmid (*SHM7/MSA1* promoter-GUS), and the effector plasmids (pH35GS-SLIM1 and pH35GS-Δ57) were introduced into the *A. tumefaciens* strains LBA4404 and GV3101, respectively. After over-night growth of *A. tumefaciens* cultures at 28 °C, the cells were collected by centrifugation and resuspended in water to an OD600 = 0.1 (reporter plasmid) or 0.4 (effector plasmids). The same aliquots of resuspended cells were mixed and infiltrated to three well-expanded leaves of 5-week-old *N. benthamiana* plants using 2 mL needleless syringes. Samples were collected for protein extractions 72 h after leaf inoculation and used for the quantitative GUS assay. The three leaf discs (8 mm diameter) collected from three different leaves were extracted with buffer (100 mM sodium phosphate pH 8.0, 10 mM EDTA, 14 mM ß-mercaptoethanol, 0.1% Triton X-100) and centrifuged at 11,000× *g* for 5 min at 4 °C. Soluble protein level was quantified using the Bradford method [30]. GUS activity was measured as the absorbance at 415 nm after mixing 10 µL of extract and 140 µL of the extraction buffer containing 1 mM PNPG (*p*-nitrophenyl-ß-D-glucuronide) and incubating the mixture at 37 °C for 30 min. The data were normalized with the absorbance of the sample without PNPG. The activity was calculated as µmol processed substrate mg^−1^ total soluble protein min^−1^.

### 3.7. Statistical Analysis

The student’s *t*-test (two-tailed) was used for pairwise comparisons between the parental plants and *slim1-2* mutants, between the SLIM1/*slim1-2* and ΔC57/*slim1-2* lines, or between the two conditions in a leaf described in Figure 3, Figure 4, Figure 5 and Figure 6. The Dunnet’s *t*-test (two-tailed) was used for multiple comparisons between the SLIM1/*slim1-2* and ΔC57/*slim1-2* or ΔC105/*slim1-2* in Figure 3. Significant differences under the same treatment were shown with asterisks (** *p* < 0.01, * 0.01 ≤ *p* < 0.05).

## Figures and Tables

**Figure 1 plants-11-02595-f001:**
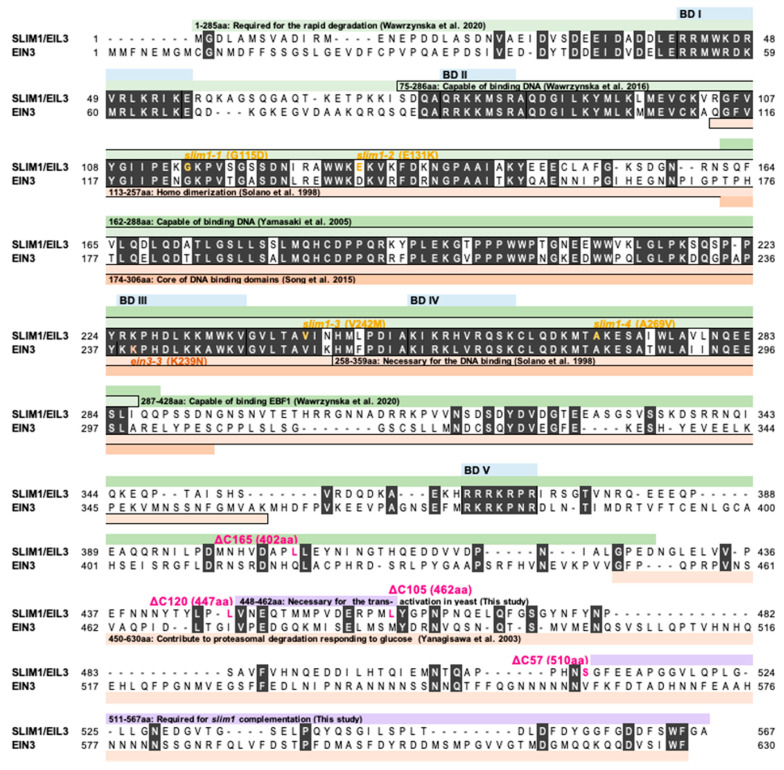
Alignment of SLIM1 and EIN3 proteins in *Arabidopsis thaliana*. Alignment of full protein sequences was performed by the ClustalW program at the DNA Data Bank of Japan (DDBJ) (http://www.ddbj.nig.ac.jp/search/clustalw-j.html, accessed on 8 May 2017). Amino acid residues conserved between SLIM1 and EIN3 are shown with white characters and dark gray background. Predicted DNA binding domains (BD I to BD V) are shown with pale blue bars. Yellow and brown characters indicate the positions of *slim1* mutations (*slim1-1*, *slim1-2*, *slim1-3*, and *slim1-4*) and *ein3-3* mutation, respectively. Magenta characters show the last amino acid in each C-terminal truncated form of SLIM1. Green and orange bars describe existing knowledge about domain structures of SLIM1 and EIN3, respectively. Violet bars highlight the findings in this study.

**Figure 2 plants-11-02595-f002:**
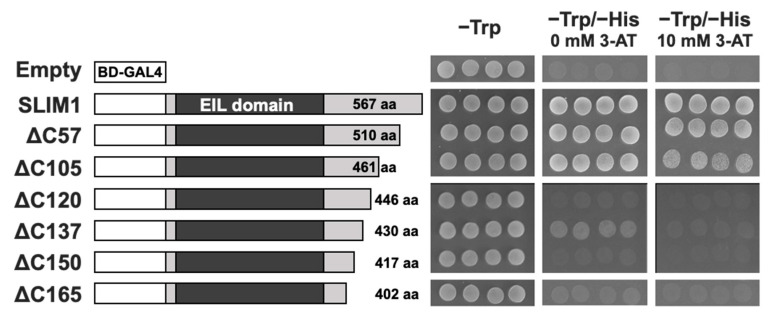
Trans-activation of *HIS3* gene expression in yeast. The left panel shows the schematic representations of SLIM1 proteins fused to the GAL4 DNA-binding domain (BD-GAL4). Empty: no insertion, SLIM1: full-length SLIM1, ΔC57, ΔC105, ΔC120, ΔC137, ΔC150, and ΔC165: C-terminal truncated forms of SLIM1 lacking the C-terminal 57, 105, 120, 137, 150, and 165 amino acids, respectively. BD-GAL4: DNA binding domain of yeast GAL4 protein. Four independent colonies obtained from each plasmid transformation of yeast strain AH109 were spotted on the minimal SD agar medium lacking Trp (−Trp) or Trp and His (−Trp/−His) with or without 10 mM 3-amino-1,2,4-triazole (3-AT). The right panel shows the yeast growth at 30 °C for 3 days after spotting.

**Figure 3 plants-11-02595-f003:**
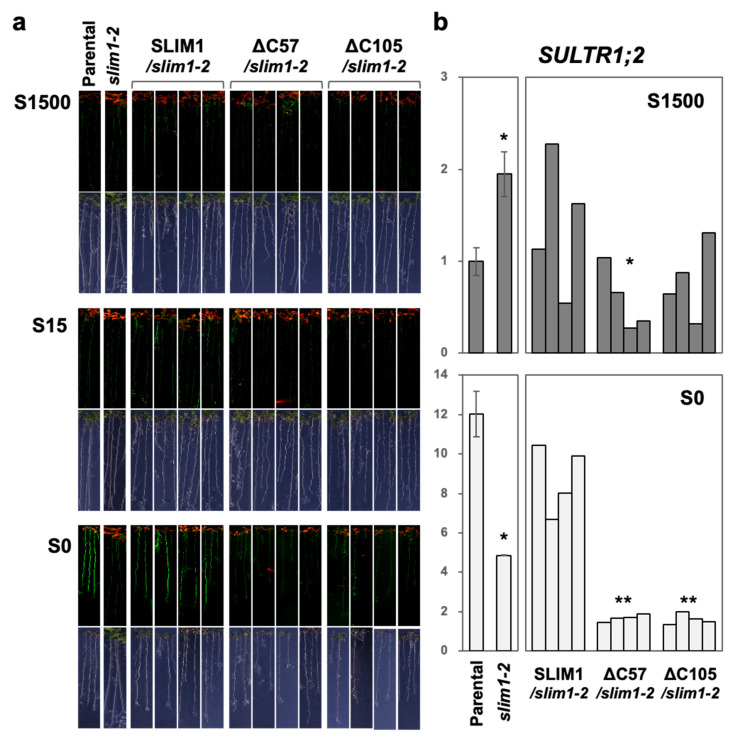
Complementation of *Arabidopsis slim1-2* by expressing the full-length and C-terminal truncated forms of *SLIM1*. The full-length *SLIM1* or the C-terminal truncated variants were expressed in *slim1-2* under CaMV 35S promoter (*SLIM1/slim1-2*, ΔC57/*slim1-2*, and ΔC105/*slim1-2*). Parental (*P_SULTR1;2_-GFP*), *slim1-2*, and four independent lines of *SLIM1/slim1-2*, ΔC57/*slim1-2*, and ΔC105/*slim1-2* were grown on S1500, S15, and S0 media for 10 days. (**a**) GFP fluorescence in plants is visualized using an image analyzer. Fluorescent images (upper panels) and bright-field images (lower panels) are shown. (**b**) Transcript levels of *SULTR1;2* in roots. The average values are indicated with error bars denoting SEM (*n* = 3) for Parental and *slim1-2* (left), and the single values are indicated for four independent transgenic lines generated for complementation with the full-length SLIM1 or the C-terminal truncated variants. Asterisks indicate significant differences between Parental and *slim1-2* determined by Student’s *t*-test (left), and between *SLIM1/slim1-2* and ΔC57/*slim1-2* or ΔC105/*slim1-2* by Dunnet’s test (right) under S1500 and S0 conditions (** *p* < 0.01, * 0.01 ≤ *p* < 0.05).

**Figure 4 plants-11-02595-f004:**
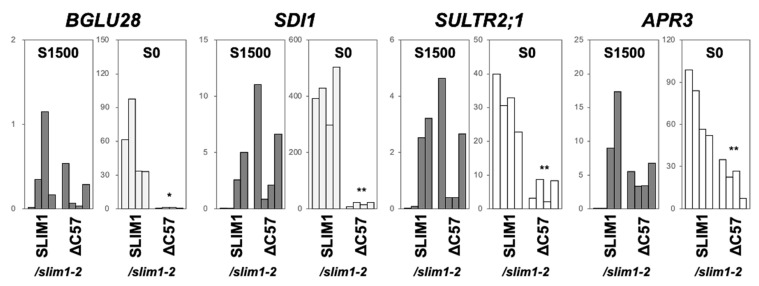
Transcript levels of −S-responsive genes in the complemented lines. Transcript levels of *BGLU28*, *SDI1*, *SULTR2;1*, and *APR3* in the roots of four independent lines of *SLIM1/slim1-2* and ΔC57/*slim1-2* were determined by qRT-PCR with the same root-derived RNA used for the *SULTR1;2* transcript expression analysis in Figure 3b. Asterisks indicate significant differences between *SLIM1/slim1-2* and ΔC57/*slim1-2* in S1500 and S0 conditions (Student’s *t*-test; ** *p* < 0.01, * 0.01 ≤ *p* < 0.05).

**Figure 5 plants-11-02595-f005:**
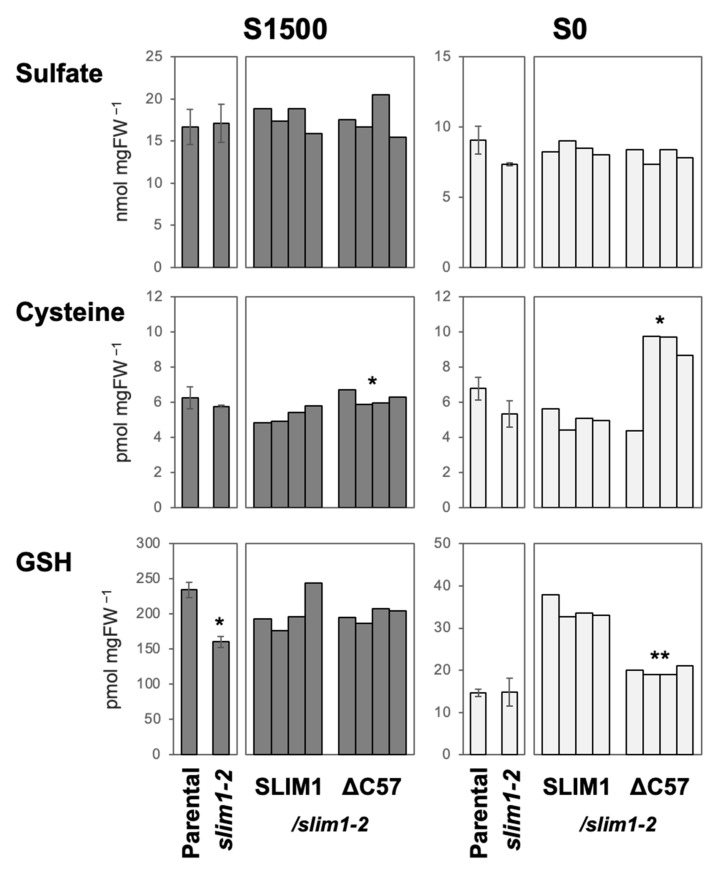
Sulfate, cysteine, and glutathione (GSH) levels in the complementation lines. Shoots of Parental (*P_SULTR1;2_-GFP*), *slim1-2*, and four independent lines of *SLIM1/slim1-2* and ΔC57/*slim1-2* grown on S1500 and S0 media for 10 days were used for the metabolite analysis. The average values are indicated with error bars denoting SEM (*n* = 3) for Parental and *slim1-2*, and the single values are indicated for four independent transgenic lines generated for complementation with the full-length SLIM1 or ΔC57. Asterisks indicate significant differences between Parental and *slim1-2* (left), and between *SLIM1/slim1-2* and ΔC57/*slim1-2* (right) under S1500 and S0 conditions (Student’s *t*-test; ** *p* < 0.01, * 0.01 ≤ *p* < 0.05).

**Figure 6 plants-11-02595-f006:**
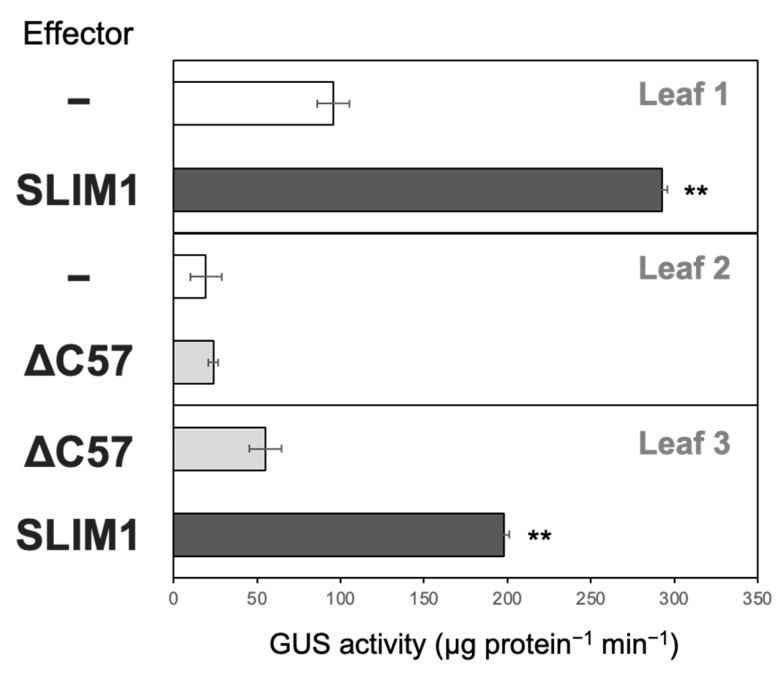
Transactivation of *SHM7/MSA1* promoter by full-length and C-terminal 57 aa truncated SLIM1. GUS activity was measured in transiently transformed *Nicotiana benthamiana* leaves incubated for 72 h after the infiltration with the reporter (*SHM7/MSA1* promoter-GUS) or the combination of the reporter and effectors (SLIM1 or ΔC57). The uidA gene in the reporter constructs was driven by the *SHM7/MSA1* promoter (569 bp upstream of ATG), and the effector expressions were driven by the CaMV 35S promoter. The mean values are indicated with error bars denoting SD (*n* = 3). Asterisks indicate significant differences between the two combinations of reporter and effector transfected in each leaf (Student’s *t*-test; ** *p* < 0.01).

## Data Availability

All data were provided within the article.

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
