# Peer review of "The C-Terminal Region of SLIM1 Transcription Factor Is Required for Sulfur Deficiency Response"

_plants, 2022, doi:10.3390/plants11192595_

Round 1
Reviewer 1 Report
In this study, Piotrowska and colleagues dissect the C-terminal domain function of the SLIM1 transcription factor in response to sulfur deficiency. The experiments performed are well designed and the conclusions are well supported by the results obtained. Although some open questions remain, for example, why Δ56 version of the C-terminal activates the expression in the transactivation assays in yeast, but it is not sufficient to complement the mutant in planta, this work presents a good characterization of the SLIM1 C-terminal domain and establish a strong foundation for future research to understand its role in planta.
I have some minor concerns:
Figure 1. When pointing to the ein3 mutation (K), there is a lack of info about the aminoacid substitution as it is pointed for slim 1-1 and slim 1-2.
Figure 2. The length of Δ106 should be 461 and not 462? And 446 for Δ121?
Line 112. Would be good too to number these aminoacids from the beginning of the protein?
Line 131. What about the other versions? Based on the results obtained from yeast, a priori, Δ121 should be tested, as Δ106 and Δ57 may complement the mutant in planta, although finally Δ57 didn’t.
Author Response
Thank you for the positive comments on the manuscript. We appreciate the helpful comments.
We have revised the manuscripts according to your suggestions. We hope this version of the manuscript meets your requirement.
>Figure 1. When pointing to the ein3 mutation (K), there is a lack of info about the amino acid substitution as it is pointed for slim 1-1 and slim 1-2.
I agree with you; thank you for the thoughtful comment. We have added the information for the ein3-3 mutation in Figure 1.
>Figure 2. The length of Δ106 should be 461 and not 462? And 446 for Δ121?
Thank you for your careful observation. After counting the number of amino acids again, we noticed that Δ106 was Δ105 and Δ121 was Δ120. We have revised them in Figures 1 to 3 and throughout the text.
>Line 112. Would be good too to number these amino acids from the beginning of the protein?
We have revised the sentence according to the suggestion.
>Line 131. What about the other versions? Based on the results obtained from yeast, a priori, Δ121 should be tested, as Δ106 and Δ57 may complement the mutant in planta, although finally Δ57 didn’t.
We did not test Δ120 (previous Δ121) but tried Δ165 and obtained a similar result with Δ105 (previous Δ106). To show that, we added a sentence in L158. In addition, we added the transactivation assay results in Figure 6 to show the C-terminal 57 aa truncation was enough to inactivate SLIM1.
Reviewer 2 Report
The presented paper the describes the role of the SLIM1 C-terminal region. Authors used the method of dissecting the region required for transactivation of the target gene expression under –S. I believe that the authors plan the research correctly and clearly define the methodology. Moreover, the research was done correctly. Both the research hypothesis and the obtained results are interesting. The most important result is that the C-terminal region of SLIM1 is indispensable to its activity under–S.
This article is worthy of attention!
Author Response
Thank you for the positive comments on the manuscript. We appreciate the helpful comments very much.
Reviewer 3 Report
My comments are as below:
Although the manuscript is well written and the information provided is good. As a comment, I would like to request the authors to add a separate discussion section and elaborate it as well so that the potential implications of their findings can be more valuable to the readers.
Author Response
Thank you for the positive comments on the manuscript. We appreciate the helpful comments in separating the Discussion section rather than the Results and Discussion.
However, there are not many discussions, and the last section (2.4. Roles of SLIM1 C-terminal region) provides some discussions separately. So we did not separate the discussion section.
We hope this version of the manuscript meets your requirement.